# Epoxyeicosatrienoic Acids and Fibrosis: Recent Insights for the Novel Therapeutic Strategies

**DOI:** 10.3390/ijms221910714

**Published:** 2021-10-03

**Authors:** Xin-Xin Guan, Dong-Ning Rao, Yan-Zhe Liu, Yong Zhou, Hui-Hui Yang

**Affiliations:** Department of Physiology, School of Basic Medical Science, Central South University, Changsha 410078, China; guanxinxin0606@163.com (X.-X.G.); 20111210088@fudan.edu.cn (D.-N.R.); liuyanzhe1007@csu.edu.cn (Y.-Z.L.); zhouyong421@csu.edu.cn (Y.Z.)

**Keywords:** EETs, renal fibrosis, cardiac fibrosis, pulmonary fibrosis, hepatic fibrosis

## Abstract

Organ fibrosis often ends in eventual organ failure and leads to high mortality. Although researchers have identified many effector cells and molecular pathways, there are few effective therapies for fibrosis to date and the underlying mechanism needs to be examined and defined further. Epoxyeicosatrienoic acids (EETs) are endogenous lipid metabolites of arachidonic acid (ARA) synthesized by cytochrome P450 (CYP) epoxygenases. EETs are rapidly metabolized primarily via the soluble epoxide hydrolase (sEH) pathway. The sEH pathway produces dihydroxyeicosatrienoic acids (DHETs), which have lower activity. Stabilized or increased EETs levels exert several protective effects, including pro-angiogenesis, anti-inflammation, anti-apoptosis, and anti-senescence. Currently, intensive investigations are being carried out on their anti-fibrotic effects in the kidney, heart, lung, and liver. The present review provides an update on how the stabilized or increased production of EETs is a reasonable theoretical basis for fibrosis treatment.

## 1. Introduction

Epoxyeicosatrienoic acids (EETs) are endogenous lipid metabolites of arachidonic acid (ARA) synthesized by cytochrome P450 (CYP) epoxygenases [1]. The main human CYP epoxygenases are CYP2C8/9/19 and CYP2J2, which are capable of converting ARA to all four EETs: 5,6-EET, 8,9-EET, 11,12-EET, and 14,15-EET (Figure 1) [2]. Although CYP epoxygenase genes are expressed in nearly all tissues, there is significant inter-tissue variability, such that EET formation is different in the various organs, and this can significantly influence the physiological response of a given tissue [3]. EETs exhibit several protective effects, such as anti-inflammation [4], regulation of vascular tone, and homeostasis [5,6]. EETs also stimulate angiogenesis and protect cells from apoptosis [7]. As the counterbalance, the pro-inflammatory terminal metabolite 20-hydroxyeicosatetraenoic acid (20-HETE) is also produced by the P450 pathway from ARA. Although inhibition of 20-HETE production appears to be beneficial in many disease states, it can only result in promising therapeutics if it could be demonstrated to stabilize the actual EET molecules [8].

EETs are rapidly metabolized via several pathways. Among them, the soluble epoxide hydrolase (sEH) pathway is dominant. The sEH pathway produces dihydroxyeicosatrienoic acids (DHETs), which display lower activity. Either using a small-molecule inhibitor or administrating a genetic knockout can reduce the activity of sEH, consequently stabilizing or increasing EET levels [9,10]. However, the deletion of the sEH gene (*Ephx2*) does not lead to higher EET levels as would be expected. Instead, the increased EET levels shift ARA metabolism to other pro-inflammatory pathways (e.g., ω-hydrolase-LOX pathways), and this counteracts the increase in EETs [11]. Other approaches, such as CYP2J2 overexpression, EET analog treatment, and exogenous EET treatment, are widely used [9,12,13,14]. Moreover, the in vivo overexpression of CYP2J2 could increase EET synthesis, and EET analogs could then lower the metabolism and improve solubility. EET analog treatment along with exogenous EET treatment could also offer corroboration for the independent effects of EETs. At any rate, the sEH inhibitor (sEHI) has advanced further toward clinical use than other metabolites. The details of sEHI have been reviewed elsewhere [15].

Tissue fibrosis formation is a complex and highly orchestrated process that is the product of a cascade of cellular and molecular responses [16], causing high morbidity and mortality worldwide [17]. The initial fibrotic response is triggered by primary damage to an organ, leading to the activation of the effector cells that drive the fibrogenic process. Activated effector cells cause the extracellular matrix (ECM) to elaborate and develop into the dynamic deposition, leading to end-organ failure [18]. Previous studies have identified many effector cells (e.g., activated fibroblasts, the senescence of parenchymal cells) [19,20,21,22], molecular pathways (e.g., transforming growth factor-beta (TGF-β) pathways) [23], and organ fibrosis is reversible in some circumstances [20]. To date, there are few effective therapies for tissue fibrosis. However, an increasing number of studies have recently indicated that EETs and EET analogs have an anti-fibrotic effect in the: kidney [24], heart [25], lung [26,27], and liver [28,29] (Table 1). This review focuses on how the stabilization or increased production of EETs leads to possible therapies for fibrosis.

## 2. EETs and Fibrosis

### 2.1. EETs and Renal Fibrosis

Compared to other organs, the kidney has a unique cellular architecture. It consists of glomeruli, tubules, interstitium, and capillaries. Injury to any of these structures will lead to the deposition of the ECM [41]. Two leading causes of renal fibrosis, hypertension, and diabetes, increase glomerular capillary pressure, which leads to proteinuria and triggers an immune response resulting in epithelial cell and interstitial fibrosis [42]; renal fibrosis is the hallmark of chronic kidney disease progression [43]. 

Lowered serum EET levels have been observed in patients with cardiorenal syndrome [44]. In rats, a decrease in EET formation promotes the development of diabetic nephropathy (DN) by increasing reactive oxygen species (ROS) production and exacerbating tubular injury [45]. EETs are also activators of peroxisome proliferator-activated receptor gamma (PPAR-γ) [46]. RB394, a dual PPAR-γ agonist/sEH inhibitor, ameliorates DN by reducing renal interstitial fibrosis [47]. These results suggest that EETs may have a reno-protective effect in chronic kidney disease, evidenced by studies using the potent sEHI and other enzymes to amplify endogenous EETs. 

In mice with unilateral ureter obstruction (UUO), raised EET concentration diminished kidney fibrosis by reducing fibroblast mobilization [24,30,31]. Activated fibroblasts can originate from various sources: directly from interstitial fibroblasts, from epithelial mesenchymal transition (EMT), or from endothelial-to-mesenchymal transition (EndMT). These processes are found to be decreased for in vitro models [24,30]. There is also a decline in the inflammatory response, oxidative stress, and cell death [31]. Moreover, it has been demonstrated that these anti-fibrotic and anti-inflammatory effects are independent of EET effects on blood pressure. EETs activate PPAR-γ, downregulate nuclear factor kappa B (NF-κB), reduce UUO-induced tubular cell death, and inhibit TGF-β1/Smad3 signaling pathways [31]. In the chronic renal failure (CRF) model, increased EET generation due to overexpression of CYP2J2 lessens collagen deposition and renal cell apoptosis via regulation of protein expression, of TGF-β1/Smads, matrix metalloproteinases (MMPs), mitogen-activated protein kinases (MAPKs), and apoptosis-related proteins. Together, these factors show a potent protective effect against apoptosis and fibrosis [32]. In stroke-prone spontaneously hypertensive rats (SHRSP), boosted EETs would protect against the pathology of fibrinoid degeneration and hypertrophy of arterioles in the kidney [48]. It has been shown that induction of heme oxygenase-1 (HO-1) by sEHI contributes to a decrease in renal fibrosis in diabetic SHR, producing increased EET levels, reduced renal collagen deposition, and fibronectin expression together with a reduction in glomerular TGF-β1 levels [33].

EET analog treatment attenuates kidney damage, which usually results in a high probability of developing renal fibrosis. These include DN, angiotensin II (Ang II)-induced hypertension, and salt-sensitive hypertension. These fibrosis preventative effects are modulated by reducing apoptosis, inflammation, and endoplasmic reticulum stress (ERS) [49,50,51,52]. Thus, EETs may be a potential anti-fibrotic therapeutic agent in renal fibrosis (Figure 2).

### 2.2. EETs and Cardiac Fibrosis

An increased risk of adverse cardiac events such as arrhythmias and sudden cardiac death is strongly correlated with fibrotic scarring [53,54]. In response to injury, the heart undergoes structural and functional remodeling, which can cause the hypertrophy of cardiac myocytes, resulting in the excessive deposition of ECM [55]. Cardiac fibroblasts are the most abundant cell type in the myocardium, and injury triggers these cells to proliferate and differentiate into myofibroblasts. This process is driven by effectors such as TGF-β1, endothelin-1, and Ang II [56].

There is increasing evidence indicating that improved EETs could diminish the fibrosis associated with cardiac hypertrophy and remodeling, as well as apoptosis and inflammation [25,57]. In thoracic aortic constriction (TAC) hearts, treatment with sEHIs was shown to result in a significant decrease in cardiac fibrosis. A significant induction of sEH gene expression was observed during Ang II-induced cardiac hypertrophy. sEH has been shown to be an essential contributor to the pathological process and is sufficient for inducing cardiac hypertrophy. All these effects can be reversed by sEHI [58]. In these studies, it was unclear whether or not sEHI exerts its cardioprotective effects through enhancing EET activity. In another study, the cardioprotective phenotype of sEH null mice was wholly abolished in the presence of EET antagonists [59]. Additionally, isoprenaline (Iso)-induced hypertrophy and fibrosis are protected against by sEHI. Those protective effects can be replicated by the addition of exogenous EETs in vitro, causing a decline in pro-fibrotic cytokines. Exogenous EETs were even shown to weaken the induction of *Ephx2* gene expression [60]. It has been reported that EETs block the fibrotic response in Ang II-infusion hearts by inhibiting Ca^2+^-calcineurin (CaN)/NFATC3 signaling [34]. The increase in EETs due to CYP2J2 overexpression is shown to reduce cardiac fibroblast activation, proliferation, migration, and secretion by inhibiting the Gα12/13/RhoA/ROCK pathway via NO/cGMP activation [35]. Additionally, there is evidence indicating that enhanced EET formation minimizes cardiomyocyte hypertrophy and remodeling [61] while also limiting collagen accumulation by inhibiting the oxidative-stress-mediated NF-κB pathway via PPAR-γ activation [62]. The decline of the TGF-β1/Smad pathway and fibroblast activity is also seen in vitro.

Furthermore, a reduction of TGF-β, connective tissue growth factor (CTGF), and procollagen gene expression caused by EETs were observed in a genetic analysis [38]. With the administration of sEHI, collagen synthesis genes such as collagen I, CTGF, and lysyl oxidase are suppressed. In contrast, *Ephx2* gene deletion improves the expression of the collagen synthesis genes and aggravates the cardiac fibrosis induced by Ang II. This is thought to be due to *Ephx2* gene disruption causing a shift in ARA metabolism, which may lead to pathological processes [11]. Increased macrophage filtration into the myocardium induced by an Iso or Ang II infusion can be suppressed by EETs, as macrophages are known to trigger the differentiation of fibroblasts into myofibroblasts [36]. Furthermore, a novel mechanism has been found that heightens EET reduction in cardiac hypertrophy and fibrosis through the activation of AMPKα2 and Akt [37]. EETs can activate the PI3K- and ATP-sensitive potassium channel (KATP) to display indirect protective effects against fibrosis [63]. In ethanol-induced cardiac fibrosis, recent results indicate that sEH inhibition prevents the activation of cardiac fibroblasts and restores their impaired autophagic flux by suppressing mTOR activation [39]. These findings demonstrate that EETs’ actions on cardiac fibrosis involve cell-signaling events that reduce apoptosis, oxidative stress, and ECM deposition (Figure 3). Although the fibrogenic genes were decreased, the treatment failed to reverse the later progression of fibrosis with no significant change in pathological results in [64]. This unsatisfactory outcome may be a result of the vasoactive properties of EETs, which sometimes play a dual role. For example, pro-angiogenic effects enhance wound repair, but they can also result in the accelerated growth of solid tumors [65].

### 2.3. EETs and Pulmonary Fibrosis

Pulmonary fibrosis (PF) is a type of chronic, progressive, and irreversible interstitial lung disease [66]. Both the inflammatory pathway and the epithelial pathway can lead to pulmonary fibrosis [67,68]. Not all types of PF involve substantial inflammation. For example, the fibrotic response of idiopathic pulmonary fibrosis (IPF) is mainly driven by abnormally activated alveolar epithelial cells (AECs). There is a vicious cycle of injury and effector-cell activation in pathogenesis. Damage to AECs initially activates pulmonary fibroblasts, which then differentiate into collagen-secreting myofibroblasts. These activated fibroblasts cause abnormal repair of the alveolar epithelium, which leads to further fibroblast activation [69]. It has been suggested that TGF-β plays a central role in the pathogenesis of PF by promoting the activation, proliferation, and differentiation of epithelial cells and myofibroblasts [70]. Researchers have been working on moving the TGF-β pathway ‘from bench to bedside’. There are also studies focusing on reducing oxidative stress and apoptosis [71]. Since most studies focus on slowing the progression of this process rather than paying attention to existing ECM deposition, we are looking for an intervention targeting matrix metalloproteinases. In this case, the aim is for the rate of degradation to overtake the rate of deposition, eventually reversing the fibrogenic process.

We have investigated the protective effects of EETs against pulmonary diseases, and recently we focused on their anti-fibrotic effects and the underlying mechanism. We used a selected sEHI (1-trifluoromethoxyphenyl-3-(1-propionylpiperidin-4-yl) urea (TPPU) and dual sEH/COX-2 inhibitor (4-(5-phenyl-3-{3-[3-(4-trifluoromethylphenyl)-ureido]-propyl}-pyrazol-1-yl)-benzenesulfonamide) (PTUPB) on bleomycin-induced PF in mice. Our results demonstrated that the inhibition of sEH attenuated inflammation, collagen deposition, and senescence [26,72], preventing bleomycin-induced PF in a mouse model. Additionally, AUDA (an sEHI) is reported to inhibit the proliferation and collagen synthesis of mouse fibroblasts and to partially reverse TGF-β1-induced α-smooth muscle actin (α-SMA) expression and EMT. These effects are partially mediated via the downregulation of the Smad3 and p38MAPK signaling pathways [27]. It has been suggested that CYP2J2 overexpression and exogenous EET treatment can attenuate TNF-α-induced apoptosis and pulmonary vascular remodeling [73], protecting against oxidative stress and apoptosis following lung ischemia/reperfusion by activating the PI3K/Akt signaling pathways [74].

Abnormal angiogenesis has always been linked with the development of fibrosis, particularly PF. In general, vascular development is firmly regulated by the coordinated expression of two signal proteins: pigment epithelium-derived factor (PEDF), a potent angiostatic cytokine; and vascular endothelial growth factor (VEGF), a potent angiogenic cytokine. In IPF, there is an increase in PEDF expression with a decrease in VEGF expression within the fibroblastic focus, which leads to a localized reduction of vascular density. It has been demonstrated that TGF-β1 stimulates PEDF expression in fibroblasts [75]. EETs are closely correlated with blood vessel regeneration. Moreover, 14,15-EET-induced angiogenesis is mostly dependent on promoting VEGF expression [76], and EETs can reverse the effects of TGF-β. These suggest that EETs might inhibit the development of PF by promoting the regional angiogenesis of fibroblastic foci. Therefore, the inhibition of EETs against inflammation, apoptosis, oxidative stress, and ECM deposition appear to be important signaling mechanisms by which EETs attenuate PF (Figure 4).

### 2.4. EETs and Hepatic Fibrosis

Hepatic fibrosis typically results from an inflammatory response by hepatocytes or biliary cells [77]. This inflammatory response is often the result of viral hepatitis, alcohol, fatty liver, and biliary tract disease and eventually leads to cirrhosis. Inflammation leads to the activation of effector cells, which results in the deposition of the ECM. Pericyte-like stellate cells were found to be the primary source of myofibroblasts during the fibrotic response in the liver [78]. Along with numerous similar molecular pathways found in other organs, the liver has a unique path involving toll-like receptor 4 (TLR4). The activation of TLR4 on the surface of stellate cells triggers cell activation and fibrogenesis [79,80].

Recent studies have shown that the pharmacological inhibition and gene deficiency of sEH attenuate carbon tetrachloride (CCl_4_)-induced hepatic fibrosis [28,29,40]. In CCl_4_-induced cirrhosis, the reduction in EET levels and the induction of sEH are reversed, and EET concentration, as well as the EET/DHET ratio, are increased following sEHI treatment. The protective effect against fibrosis is primarily due to the inhibition of HSC activation and NF-κB signaling by EETs [29]. However, in a series of studies on CCl_4_-treated mice, hepatic EET levels did not increase significantly after treatment with sEHIs. That may have been due to the slight increase in EET levels triggered by CCl_4,_ which was insufficient to counter the fibrotic process. Thus, sEH inhibition reduced not only collagen deposition but also the mRNA level of ECM components. Inhibition of sEH also altered the activity of MMPs [28]. Moreover, sEHI inhibits ERS, which could affect ECM synthesis and the progression of fibrotic diseases [28,81]. These findings demonstrate that EET actions on hepatic fibrosis involve NF-κB, ERS, and MMP as well as other cell-signaling events (Figure 5).

Although EETs can significantly attenuate inflammation in nonalcoholic fatty liver disease [82], EET levels appear to be increased in cirrhotic portal circulation. Therefore, EETs may be involved in the pathophysiology of portal hypertension [83]. These opposite results may be associated with the vasoactive properties of EETs.

## 3. Conclusions

Overall, the present review spotlights the distinctive anti-fibrotic contribution of EETs in organ fibrosis via multiple pathways and discusses the pathogenetic role of sEH and how stabilized or increased production of EETs leads to a theoretical basis for the treatment of fibrosis. Growing evidence suggests that mitochondrial dysfunction and senescence contribute to renal fibrosis, cardiac fibrosis, PF, and hepatic fibrosis [19,84,85]. More importantly, there are increasingly more studies indicating that EETs may be potential endogenous molecules for anti-aging. Therefore, the role of EETs in senescence should be validated in future studies.

## Figures and Tables

**Figure 1 ijms-22-10714-f001:**
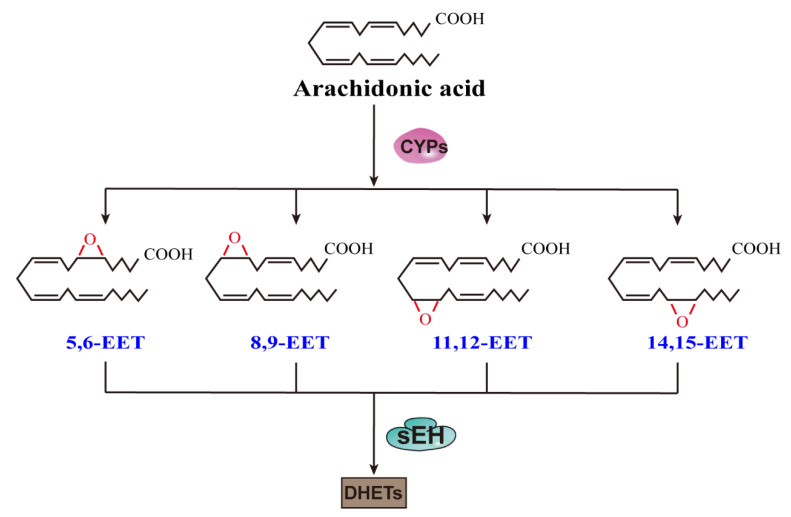
EET regioisomers synthesized from arachidonic acid by CYPs and their conversion to DHETs by sEH.

**Figure 2 ijms-22-10714-f002:**
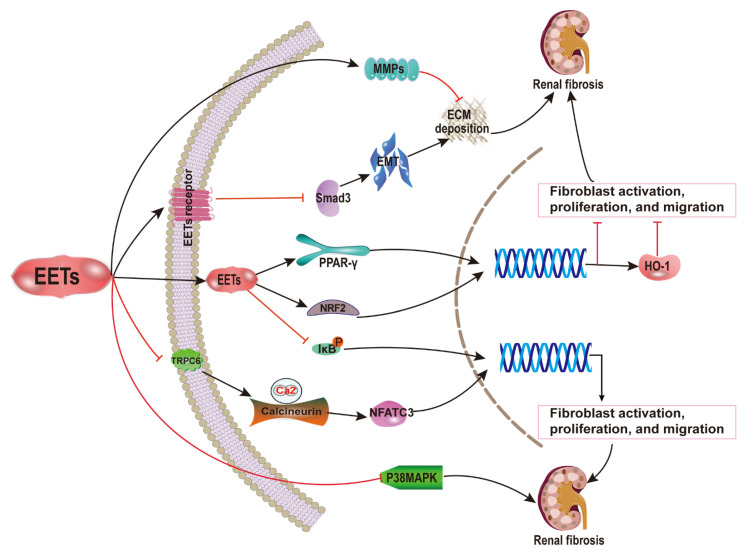
Anti-renal fibrosis mechanisms of EETs. The fibrotic preventative effects of EETs are mediated by the reduction of apoptosis, inflammation, and endoplasmic reticulum stress. EETs activate PPAR-γ, downregulate NF-κB, reduce UUO-induced tubular cell death, and inhibit TGF-β1/Smad3 signaling pathways. EETs lessen collagen deposition and renal cell apoptosis via the regulation of protein expression, including TGF-β1/Smads, MMPs, MAPKs, and apoptosis-related proteins. Thus, EETs could be a promising strategy for the clinical treatment of renal fibrosis.

**Figure 3 ijms-22-10714-f003:**
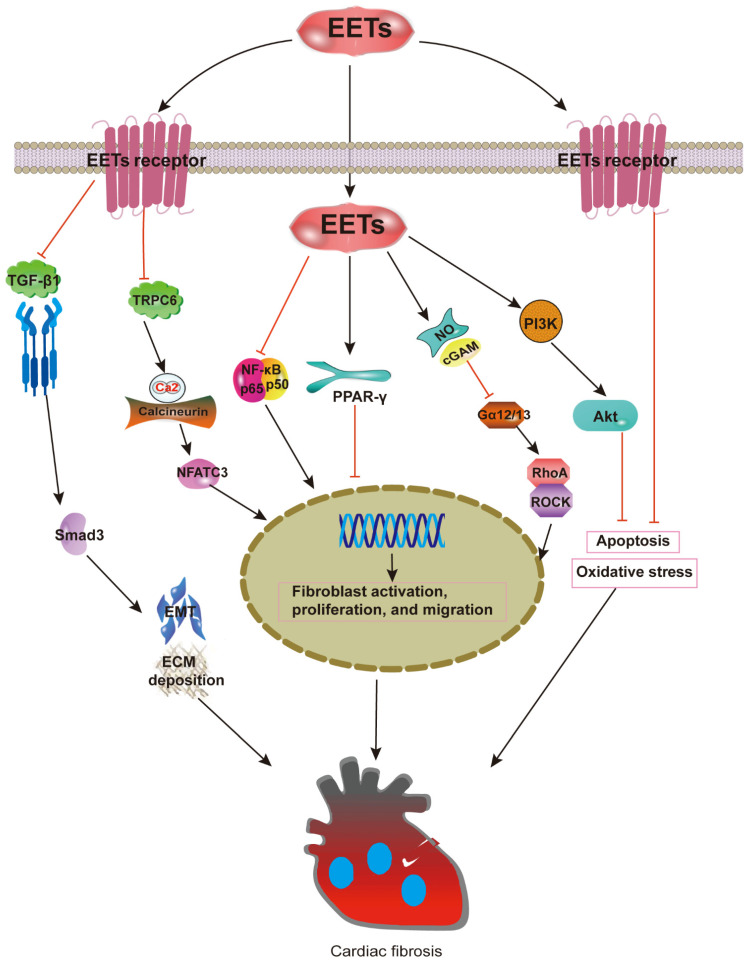
Anti-cardiac fibrosis mechanisms of EETs. EETs could diminish fibrosis associated with cardiac hypertrophy and remodeling, as well as apoptosis and inflammation. EETs block the fibrotic response in Ang II-infusion hearts by inhibiting CaN/NFATC3 signaling. The increase of EETs is shown to reduce cardiac fibroblast activation, proliferation, migration, and secretion capacity by inhibiting the Gα12/13/RhoA/ROCK pathway via NO/cGMP activation. Additionally, enhanced EET formation minimizes cardiomyocyte hypertrophy and remodeling, while limiting collagen accumulation through the inhibition of the oxidative-stress-mediated NF-κB pathway via PPAR-γ activation.

**Figure 4 ijms-22-10714-f004:**
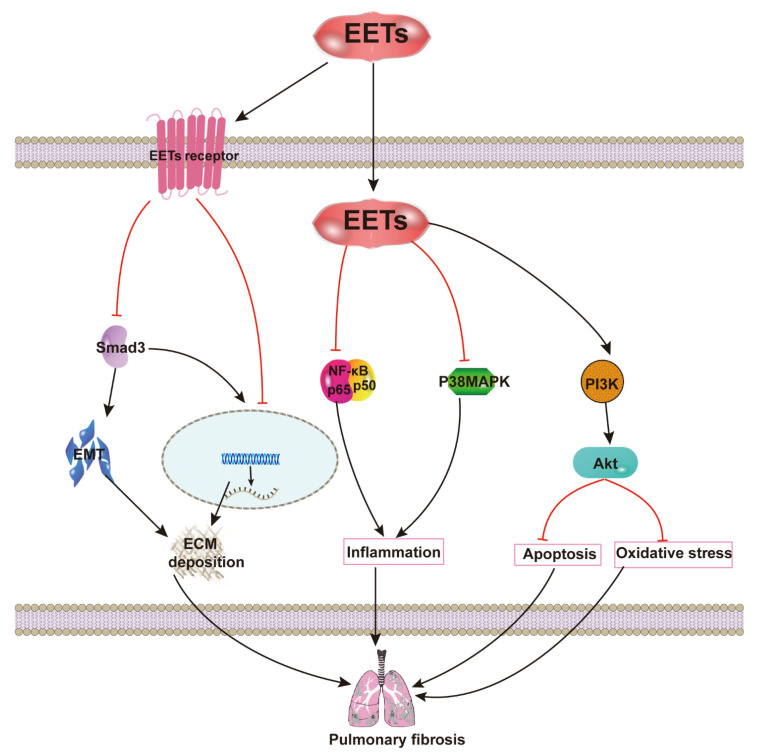
Anti-pulmonary fibrosis mechanisms of EETs. EETs inhibit the proliferation and collagen synthesis of mouse fibroblasts, partially reversing TGF-β1-induced α-SMA expression and EMT. These effects are found to be partially mediated via downregulation of the Smad3 and p38MAPK signaling pathways. Exogenous EET treatment attenuates TNF-α-induced apoptosis and pulmonary vascular remodeling, protecting against oxidative stress and apoptosis following lung ischemia/reperfusion by activating the PI3K/Akt signaling pathways.

**Figure 5 ijms-22-10714-f005:**
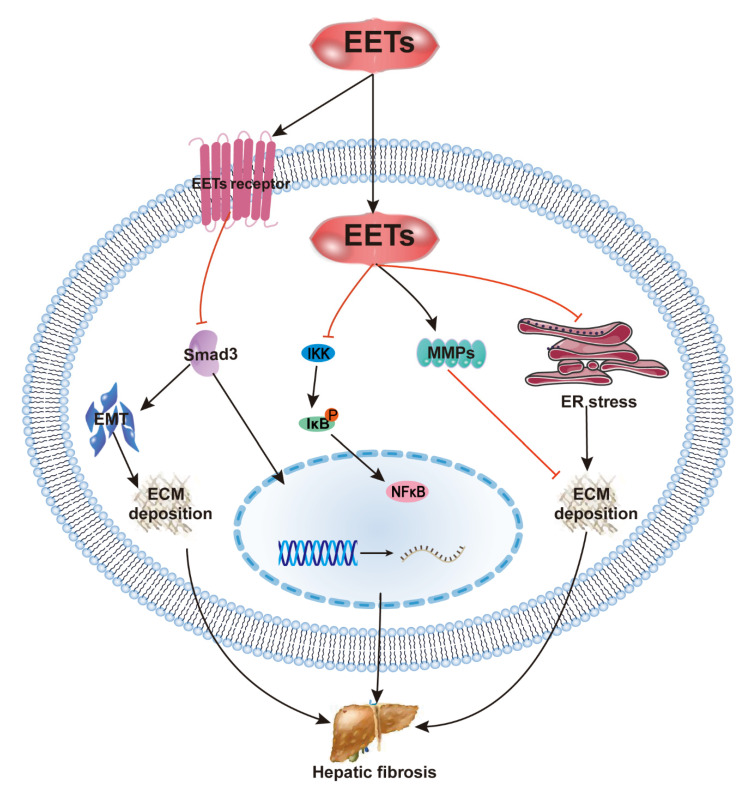
Anti-hepatic fibrosis mechanisms of EETs. The protective effect against hepatic fibrosis is mostly due to inhibition of HSC activation and NF-κB signaling by EETs, sEH inhibition reduced collagen deposition, and the mRNA level of the ECM components. EETs alter the activity of MMPs and inhibit ER stress, which could affect ECM synthesis and the progression of hepatic fibrosis.

**Table 1 ijms-22-10714-t001:** Direct anti-fibrotic effects and mechanisms of EETs.

	Model/Cell	Method	Effect and Mechanism
**Kidney**	UUO [24,30,31]	Ephx2^−/−^ sEHI (tAUCB, t-TUCB, AUDA) EET-A Exogenous 14,15-EET	Reduce fibrotic response of fibroblasts, collagen deposition, inflammation, and oxidative stress, and prevent EndMT and EMT Inhibit PPAR1, NF-κB, and TGF-β1/Smad3 signaling Activate PPAR-γ
	5/6-Nx [32]	CYP2J2 overexpression (rAAV-CYP2J2)	Reduce collagen deposition, apoptosis, and renal dysfunction Inhibit TGF-β1/SMAD signaling, MAPKs, and JNKs Modulate MMPs Regulate apoptosis-related proteins
	Diabetic SHR [33]	sEHI (t-AUCB) Exogenous 11,12-EET	Reduce collagen deposition and inflammation Reduce TGF-β1, NF-κB, MCP-1, and IL-17 Induce HO-1
**Heart**	Ang II infusion hearts [11,34,35,36,37]	CYP2J2 overexpression (rAAV-CYP2J2) aMHC-CYP2J2-Tr Exogenous 11,12-EET and 14,15-EET sEHI (TUPS, TPPU)	Reduce fibrotic response of fibroblasts, collagen deposition, hypertrophy, remodeling, and oxidative stress and inflammation Inhibit NF-κB, TGF-β1/SMAD, Gα12/13/RhoA/ROCK, and Ca^2+^-calcineurin/NFATc3 signaling Decrease pro-fibrotic cytokines (LOX, CTGF) Activate NO/cGMP signaling and PPAR-γ Enhance Akt1 nuclear translocation through interaction with AMPKα2β2γ1
	Post-MI [38]	sEHI (GSK2188931B)	Reduce fibrotic response of fibroblasts, collagen deposition Inhibit gene transcription of TGFβ, CTGF, and procollagen I
	Iso infusion hearts [36]	CYP2J2 overexpression (aMHC-CYP2J2-Tr) 14,15-EET	Reduce fibroblasts activation and macrophage infiltration Inhibit NF-κB and TGF-β1
	EtOH [39]	sEHI (TPPU)	Reduce fibroblasts activation and restore autophagic flux Inhibit mTOR activation
**Lung**	BLM [26,27]	sEHI (TPPU, AUDA)	Reduce fibrotic response of fibroblasts, collagen deposition, and inflammation and prevent EMT Inhibit TGF-β1/Smad3, Smad3/p38 signaling
**Liver**	CCl_4_ [28,29,40]	sEHI (TPPU, t-TUCB) sEH gene disruption	Reduce collagen deposition Inhibit NF-κB signaling, HSC activation, and gene transcription of ECM components Reduce ER stress Modulate MMPs

## Data Availability

Not applicable.

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
