# Peer review of "Epoxyeicosatrienoic Acids and Fibrosis: Recent Insights for the Novel Therapeutic Strategies"

_ijms, 2021, doi:10.3390/ijms221910714_

Round 1
Reviewer 1 Report
Authors have dicussed anti-fibrosis property of EETs in kindeys, heart, lung, and liver.
For mechansims as indicated in all 4 figures, authors show that EETs enter cells and activate a number of protective pathways. Authors failed to discuss the possibility that EETs can interact with EET receptor on plasma membrane.
Authors also failed to compare the efficacy of epoxy groups at different carbon (5-6, 8-9, 11-12, 14-15) on arachidonic acid.
Please add abbreviation of EETs in Introduction.
Please indicate Figure 1, 2, 3, and 4 in texts.
In Figure 1 legend, 'achieves' should be 'is achieved'.
Reviewer 2 Report
I have found this review by Xin-Xin Guan and colleagues interesting and easy to follow. To make this review more reader-friendly, I suggest the authors including a table for each described fibrotic condition, summarising the beneficial effects (antiinflammatory) of EETs in these organs.
